# Heterogeneity in tumor chromatin-doxorubicin binding revealed by in vivo fluorescence lifetime imaging confocal endomicroscopy

Hugh Sparks [1,2], Hiroshi Kondo[1], Steven Hooper[1], Ian Munro[2], Gordon Kennedy[2], Christopher Dunsby[2], Paul French[2] & Erik Sahai [1]

We present an approach to quantify drug–target engagement using in vivo fluorescence endomicroscopy, validated with in vitro measurements. Doxorubicin binding to chromatin changes the fluorescence lifetime of histone-GFP fusions that we measure in vivo at single-cell resolution using a confocal laparo/endomicroscope. We measure both intra- and inter-tumor heterogeneity in doxorubicin chromatin engagement in a model of peritoneal metastasis of ovarian cancer, revealing striking variation in the efficacy of doxorubicin–chromatin binding depending on intra-peritoneal or intravenous delivery. Further, we observe significant variations in doxorubicin–chromatin binding between different metastases in the same mouse and between different regions of the same metastasis. The quantitative nature of fluorescence lifetime imaging enables direct comparison of drug–target engagement for different drug delivery routes and between in vitro and in vivo experiments. This uncovers different rates of cell killing for the same level of doxorubicin binding in vitro and in vivo.

[1] Tumor Cell Biology Laboratory, The Francis Crick Institute, London NW1 1AT, UK. [2] Photonics Group, Physics Department, Imperial College London, South Kensington Campus, London SW7 2AZ, UK. Correspondence and requests for materials should be addressed to C.D. (email: christopher.dunsby@imperial.ac.uk) or to P.F. (email: paul.french@imperial.ac.uk) or to E.S. (email: erik.sahai@crick.ac.uk)

The failure of chemotherapy to effectively target all cancer cells within a tumor is a major problem in cancer treatment. In many cases, uneven drug distribution within tissue is a significant contributing factor in the heterogeneity in therapeutic response[1–3]. Further hindrances to chemotherapy include signals from the tumor microenvironment that reduce the efficacy of the therapeutic agent[4]. To study these problems in detail requires methods for accurately determining drug–target engagement in vivo. Typically, analytical chemistry techniques are used to determine drug concentration in plasma and tissue (pharmacokinetics) and evaluation of changes in a biomarker downstream of the drug–target (pharmacodynamics). Analytical chemistry can provide highly accurate measurements, but the sampling of plasma or homogenization of the tissue involved means that inter-cellular variation in drug–target engagement is not measured and intracellular drug–target binding is not quantified. Biomarker analysis can investigate inter-cellular variation in response if histochemical methods are used; however, it is not a direct measure of target engagement and downstream biology and feedback mechanisms can mean that biomarkers do not always reflect drug–target binding. To overcome these issues, methods that enable in situ visualization and quantification of drug–target binding are required.

Intravital fluorescence microscopy is a powerful method to investigate heterogeneity in cancer cell behavior and state in situ. The engineering of fluorescent reporters, read out using a range of quantitative microscopy techniques[5], can provide information about the activity of numerous kinases and transcription factors[6]. Further, intrinsic fluorescence of drugs or fluorescent labeling of drugs or biological therapeutics enables their distribution to be monitored. Here we develop intravital fluorescence microscopy to read out drug–target engagement by exploiting the quenching of emission from an intracellular donor fluorophore through Förster resonance energy transfer (FRET)[7,8] to a second fluorophore (in this case a fluorescent drug) that comes into close proximity of the donor. The rate at which FRET quenches emission is inversely proportional to the sixth power of the distance between the donor and acceptor fluorophores and is typically only significant over ~10 nm. Since FRET provides an additional means for the donor fluorophores to lose energy, it results in a decrease in their fluorescence lifetime, which is the average time a fluorophore stays in its excited state. Thus, drug–target engagement can be detected and quantified through measurement of the fluorescence lifetime of a fluorophore labeling the target if the drug has spectroscopic properties suitable for FRET. Fluorescence lifetime imaging (FLIM), which entails measuring the fluorescence lifetime for every pixel in a field of view, can be used to quantify FRET, and therefore provide a map of drug–target engagement.

Fluorescence lifetime measurements are particularly useful for in vivo application since the read out does not depend on the fluorophore concentration, emission intensity, or the relative intensity of signals in different spectral channels. Thus, fluorescence lifetime readouts are insensitive to the (spectral) attenuation properties of the sample (inner filter effect) and can be directly compared between instruments and between different samples. Furthermore, if the donor fluorescence signal can be fitted to a suitable complex exponential decay model, it is possible to obtain the population fraction of donor fluorophores that are undergoing FRET. While fitting a fluorescence decay profile to a single exponential decay model (with ~10% accuracy) requires 100 s of photons, fitting to a complex decay profile requires many 1000 s of photons[9]. For FRET experiments, it is common to fit the donor fluorescence decay profiles to a double exponential decay model corresponding to a mixture of signals from FRETing and non-FRETing donor fluorophores. If all the non-FRETing donor fluorophores are assumed to have a constant (spatially invariant) fluorescence lifetime and all the FRETing donor fluorophores are assumed to decay with a (shorter) fluorescence lifetime that is also constant across the field of view (or dataset), then complex fluorescence decay profiles can be analyzed globally, such that lifetime values can be obtained with only 100s of photons detected per pixel[10–12].

Fluorescence lifetime can be measured with a number of techniques[5,8], and in this work we utilize ultrashort pulsed excitation with time-correlated single photon-counting (TCSPC) detection[13]. This entails exciting the sample with a sufficiently low intensity such that less than one photon per excitation pulse is detected and recording the arrival time of each detected photon following the pulsed excitation. TCSPC is shot-noise limited and is conveniently implemented in laser-scanning microscopes, where each detected photon is associated with a specific image pixel. We employ an optical fiber-based laser-scanning confocal endomicroscope that is compatible with pulsed laser excitation and TCSPC acquisition and is 2.6 mm diameter at the distal end. This instrument thus enables confocal FLIM endomicroscopy and laparoscopy[14] that is physically compatible with animal disease models and it has previously been applied to read out FRET in vitro[15].

Doxorubicin[16] is an anthracycline and its mechanism of action in cancer therapy is intercalation with DNA. This hinders DNA replication and affects proliferating cells more than quiescent ones, thereby preferentially affecting cancer cells. Although doxorubicin and related molecules are routinely used in chemotherapy, including for ovarian cancer, they often only lead to partial responses with significant disease remaining[17,18]. This lack of complete efficacy is common across chemotherapies and implies either sub-optimal drug–target engagement or resistance mechanisms. One notable property of doxorubicin is its intrinsic absorption and fluorescence in the visible spectrum[19]. Although its quantum efficiency is low relative to standard fluorophores used for imaging, it has been used to study intracellular processes using fluorescence microscopy, e.g.,[20,21] and can serve as an acceptor for FRET from GFP[22]. In this study, we indirectly label DNA via the expression of EGFP fused to Histone H2B and exploit the previously reported[22] quenching of its emission by FRET to doxorubicin (DOX). This reveals significant variations in drug–target engagement depending on the delivery route. Further, considerable inter- and intra-tumor heterogeneity in drug engagement is observed that is likely to contribute to the sub-optimal efficacy of doxorubicin. Taken together, our data illustrate the utility of our approach for studying drug–target binding in pre-clinical models, including comparisons of drug delivery mechanisms and studies of heterogeneous and sub-optimal chemotherapy responses.

## Results

**Fluorescence lifetime imaging with a confocal endomicroscope.** To study molecular processes in vivo, we have developed a laser-scanning FLIM endomicroscope built around a commercial (Mauna Kea Technologies, Cellvizio®) confocal endomicroscope that acquires images through a small diameter fiber-optic bundle (Fig. 1a)[14]. Here we applied this device to make in vivo fluorescence lifetime measurements at internal locations in murine pre-clinical cancer models. Briefly, for time-resolved measurements of fluorescence, a pulsed Ti:Sapphire laser operating at 80 MHz repetition rate was frequency-doubled to provide excitation pulses at 488 nm with pulse widths in the 10 to 100 picosecond range after propagating through the fiber-optic bundle. This pulsed excitation light was introduced into the confocal scanning unit of the Cellvizio® system and scanned across the proximal face of the fiber-optic bundle containing ~30,000 optical fiber cores. A

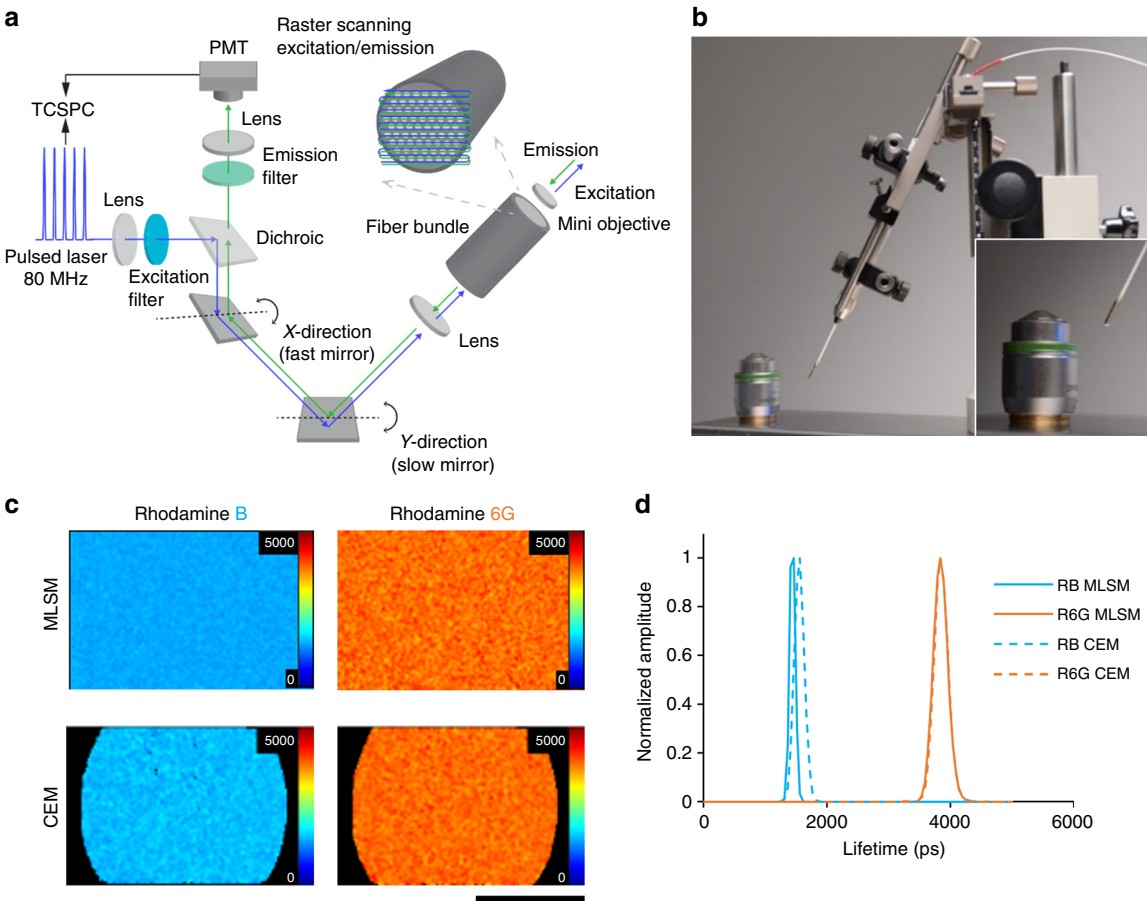

**Fig. 1** Confocal micro-endoscope system. **a** Diagram shows the confocal endomicroscope (CEM) system. For time-domain FLIM, a Ti:Sapphire laser operating at 80 MHz is frequency-doubled to generate excitation light at 488 nm. A 720 nm short pass filter rejects the infrared fundamental wavelength at 976 nm. The excitation is introduced into the confocal excitation/imaging path via a 505 nm long pass dichroic beamsplitter and directed to a resonant scanner for rapid line scanning and a galvanometric mirror for frame scanning. The scanned excitation laser beam is focused onto a coherent fiber-optic bundle such that it sequentially addresses single cores in the fiber-optic bundle, which has a diameter of ~1 mm with ~30,000 cores. At the distal tip of the coherent fiber-optic bundle, a mini-objective lens of 2.6 mm diameter images each illuminated fiber core onto the sample plane and the fluorescence signal from the sample is imaged back onto the same core of the distal end of the fiber-optic bundle. This implements optically sectioned confocal imaging with the core serving as the confocal pinhole. At the proximal end of the fiber-optic bundle the fluorescence signal is descanned and focused onto a confocal aperture in front of a photon-counting photomultiplier. Photodetection events are recorded via a TCSPC card (B&H SPC830) that associates a photon arrival time relative to the phase of the scanning unit mirrors and arrival of the excitation laser pulses. Thus, histograms of photon arrival times are assigned to each image pixel and subsequently used to generate a fluorescence lifetime image. TCSPC count rates are $\leq 10^{6}$ Hz. **b** Photograph of the confocal endomicroscope alongside a 20× multiphoton microscope objective (Carl Zeiss, objective Plan-Apochromat 20× /0.8, M27, model# 420650-9901-000), showing their relative size. **c** Fluorescence lifetime images of aqueous solutions (10 µM) of Rhodamine B (RB) and Rhodamine 6G (R6G) made using a conventional TCSPC FLIM microscope (MLSM) (Zeiss LSM-780 NDD multiphoton FLIM) and the CEM. The FLIM images are uniform as expected from a homogenous sample. **d** Fluorescence lifetime histograms corresponding to the FLIM images shown in **c**. The solid and dashed lines represent the MLSM and CEM FLIM data, respectively

miniature microscope objective on the distal tip of the fiber-optic bundle imaged the excitation light from an individual core into the sample and imaged the resulting fluorescence back to the same core. The EGFP fluorescence excited at 488 nm was passed through a 520–550 nm emission filter and detected using a photomultiplier connected to a TCSPC system. The overall diameter of the fiber-optic probe was 2.6 mm (Fig. 1b), making it well-suited to imaging internal body locations. For simplicity, we term this device a confocal endomicroscope (CEM), even though peritoneal cavity imaging is usually termed laparoscopy. A full description of the device is included in the extended Methods.

We benchmarked the FLIM capability of the CEM on solutions of rhodamine dyes with varying lifetimes. Figure 1c shows that the CEM measured the fluorescence lifetime of Rhodamine B and 6G as ~1600 ps and 4000 ps, respectively. These values closely match measurements made using a conventional objective lens-based multiphoton laser-scanning microscope (LSM) with TCSPC detection system (Carl Zeiss, LSM780 and Becker & Hickl, Simple Tau 150) (Fig. 1d) and with previously published values[23].

**FRET between chromatin-bound GFP and doxorubicin.** The previously reported ability of doxorubicin to absorb light via resonance energy transfer from GFP[22] led us to speculate that it may modulate the fluorescence lifetime of EGFP expressed in cells. We surmised that this phenomenon would indicate when EGFP and doxorubicin were within a few nanometers, as required for FRET. Given that doxorubicin exerts its action through binding to DNA that is packed into chromatin in the nucleus of mammalian

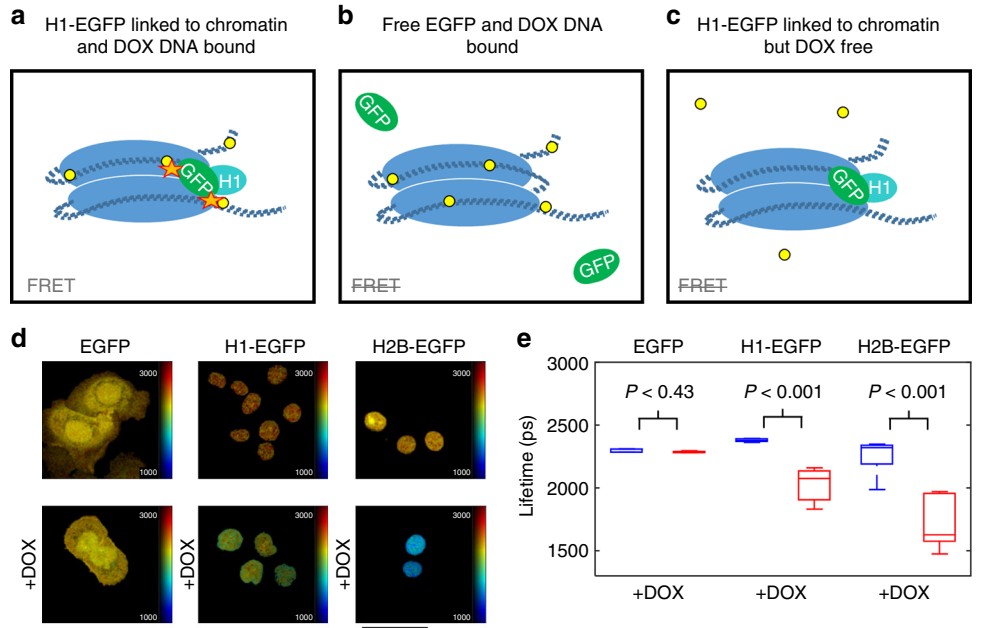

**Fig. 2** In vitro fluorescence lifetime measurement of doxorubicin binding to chromatin. **a–c** diagrams of possible interactions between doxorubicin and H1-EGFP chromatin. **a** doxorubicin (yellow dots) intercalation into DNA (blue regions represent the core histone optima and dark blue dashed lines represent DNA) brings it close to histone H1-EGFP, leading to energy transfer (orange stars). **b** GFP is not fused to histone H1 and is therefore not sufficiently close to DNA bound doxorubicin for energy transfer to occur. **c** doxorubicin is not binding DNA and therefore no energy transfer with histone H1-EGFP occurs. **d** in vitro multiphoton LSM fluorescence lifetime images of IGROV-1 cells transfected with the indicated constructs and exposed to 9 µM doxorubicin (scale bar is 210 µm). **e** Box plots of mean fluorescence lifetime values per cell corresponding to fluorescence lifetime images in **d**. Red and blue box plots represent measurements with and without incubation with doxorubicin for 3 h, respectively, for each construct. On each box plot, the central mark indicates the median, and the bottom and top edges of the box indicate the interquartile range (IQR). The box plot whiskers represent either 1.5 times the IQR or the maximum/minimum data point if they are within 1.5 times the IQR. For each construct, the data represented by the red and blue box plots were compared with an unpaired, unequal variances two-tailed t-test and the associated P-values are shown above each pair of box plots as described in the Methods section. With $n = 1$ biological replicates, the number of cells imaged for each expression type with & without (w&wo) DOX was 6&10 for cells expressing free EGFP, 12&30 for cells expressing H1-EGFP, and 8&6 for cells expressing H2-EGFP. Imaged cells for each construct expression and with&without DOX combination were sampled from a single well of a 6-well plate and all six combinations were seeded on the same 6-well plate

cells, we sought to tether EGFP to chromatin (Fig. 2a, b, c) such that a reduction in EGFP lifetime due to FRET would report doxorubicin binding. This was accomplished by fusing EGFP to protein components of chromatin, histones H1 and H2B. Figure 2d shows H1-EGFP and H2B-EGFP fusions localized to the nucleus of IGROV-1 ovarian cancer cells. This contrasts with the broader cellular distribution of free EGFP expressed throughout the cell. The fluorescence lifetime of EGFP was measured using the multiphoton LSM with two-photon excitation at 900 nm and a 465–495 nm bandpass emission filter that captures the shorter end of the GFP emission spectrum and none of the doxorubicin spectrum (Supplementary Figure 1 and Supplementary Table 1). All EGFP proteins exhibited a fluorescence lifetime in the expected 2300–2400 ps range, with H1-EGFP having a marginally longer lifetime. Crucially, when doxorubicin was added, the mean fluorescence lifetime of both H1-EGFP and H2B-EGFP decreased significantly (Fig. 2e). Equivalent results were obtained in fixed and unfixed samples. The fluorescence lifetime of soluble EGFP that was not attached to chromatin was unaffected by doxorubicin, even in the nucleus. This indicates that doxorubicin bound to DNA modulates the lifetime of chromatin-tethered GFP. The use of a 465–495 nm bandpass filter enables us to exclude the possibility of light emitted by doxorubicin from interfering with the results (Supplementary Figure 1).

**FLIM endomicroscope can monitor doxorubicin cellular uptake.** We tested whether the CEM would measure similar

changes in fluorescence lifetime to the multiphoton LSM TCSPC microscope system. IGROV-1 ovarian cancer cells stably expressing H1-EGFP were treated with increasing concentrations of doxorubicin, ranging from 180 nM to 18 µM. Figure 3 shows that similar trends were obtained for CEM and LSM lifetime measurements. The 465–495 nm emission filter used for the LSM measurements (Fig. 3a, c) was not suitable for the CEM, as it employs excitation at 488 nm. Shorter excitation wavelengths were not practical in the CEM since they would excite unwanted background fluorescence in the fiber-optic glass[24]. Instead, a 520–550 nm bandpass emission filter was used in the CEM to block the excitation light and unwanted background fluorescence (Supplementary Figure 1). With this in place, the CEM device measured a decrease in fluorescence lifetime when cells are treated with doxorubicin. Figure 3a, b show comparison data acquired with the CEM and multiphoton LSM and quantification of this data is shown in Fig. 3c, d, e. However, the changes in mean fluorescent lifetime measured with increasing concentrations of doxorubicin with the CEM were quantitatively different compared to those measured using multiphoton LSM; this was the case whether a single (Fig. 3c) or double exponential fit (Fig. 3d) was attempted. This is likely due to doxorubicin emission making a minor contribution to the CEM measurements, but not to the multiphoton LSM measurements (made with the shorter wavelength emission filter).

As discussed in the Supplementary Note 1, we undertook an in vitro study to quantify the contribution of doxorubicin emission in the CEM detection channel and its impact on the

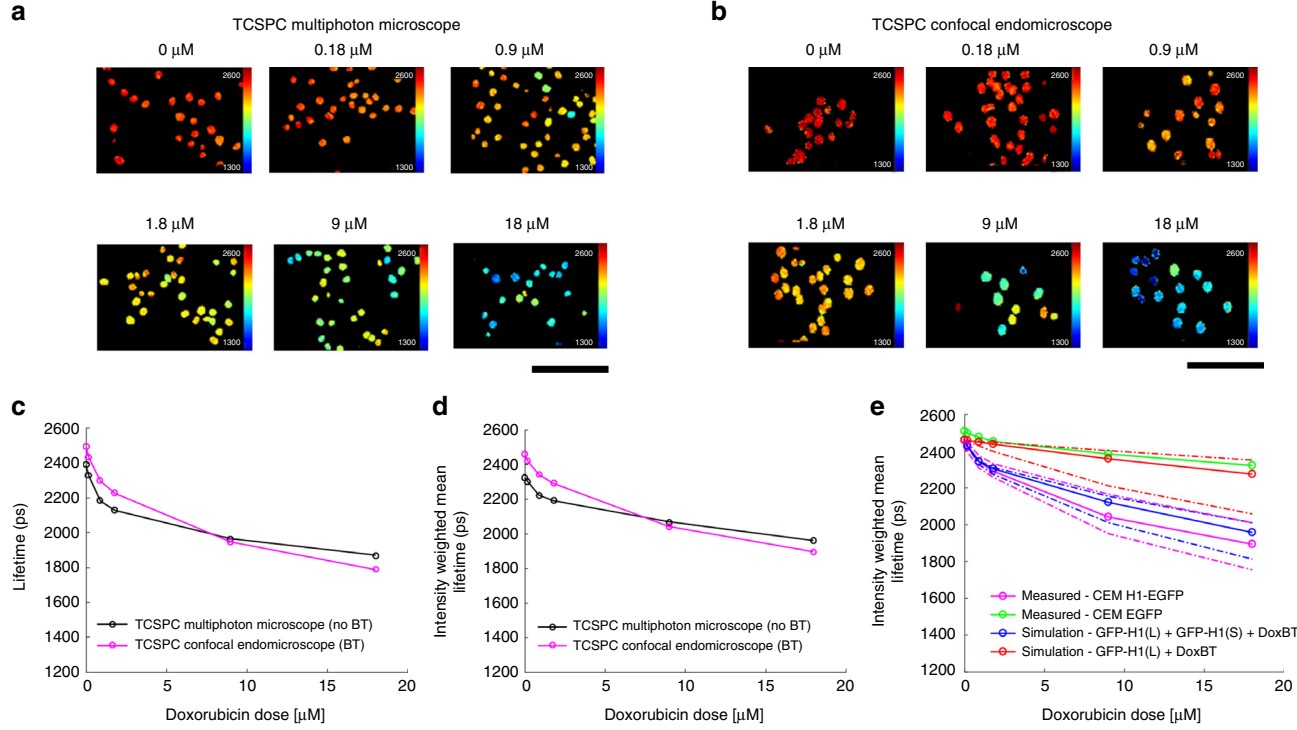

**Fig. 3** In vitro fluorescence lifetime measurement of GFP bound to chromatin for range of doxorubicin concentrations. **a** Exemplar fluorescence lifetime images of IGROV-1 cells with stable expression of histone H1-EGFP treated with a range of doxorubicin concentrations (0, 0.18, 0.9, 1.8, 9, 18 μM) for 3 h then fixed, washed and imaged with the multiphoton LSM (scale bar is 120 μm). For each concentration 3 or more fields-of-view (containing a total of >100 nuclei) were acquired. **b** Exemplar fluorescence lifetime images of the same samples as in **a** but acquired with the CEM (scale bar is 120 μm). In **a** and **b** FLIM data was fitted to a single exponential decay model (as discussed in the Methods section). **c** Plot shows the mean lifetime of segmented nuclei averaged over the mean fluorescence lifetimes per segmented nucleus (i.e., average of values from pixel-wise fitting to a single exponential decay model within a single nucleus) for each doxorubicin concentration; **d** plot shows the intensity-weighted mean H1-EGFP fluorescence lifetimes obtained by global fitting across the entire dataset (as discussed in the Methods section) to a double exponential decay model with the long lifetime component fixed to the value obtained from cells expressing H1-EGFP when not treated with doxorubicin. For **c** and **d** black and magenta lines and circles represent the multiphoton LSM and CEM data, respectively. **e** plot shows only the globally fitted intensity-weighted mean H1-EGFP lifetime for the experimental CEM FLIM data (magenta lines and circles) together with simulations of the measured lifetime dependence on doxorubicin concentration (as discussed in Supplementary Note 1) assuming no FRET but a bleed through (BT) contribution from doxorubicin fluorescence (red line and circles) and H1-EGFP FRET with the BT contribution from doxorubicin fluorescence (blue line and circles). The solid blue and red lines represent the simulated intensity-weighted mean lifetime values based on the median cell-wise measured contribution from doxorubicin fluorescence in the CEM (see Supplementary Note 1). The dashed red and blue lines indicate simulated values based on the 25th and 75th percentiles of the measured contribution from doxorubicin fluorescence. The magenta and green lines (measured intensity-weighted mean lifetime values) show the median (solid lines and circles) and 25th and 75th (dashed lines) percentile values of the distribution across all H1-GFP expressing cells and GFP expressing cells, respectively

measured EGFP fluorescence lifetime. Using confocal LSM and spectral unmixing of the fluorescence excited at 488 nm from IGROV-1 H1-EGFP nuclei treated with known concentrations of doxorubicin, we calculated the relative contributions from the EGFP and doxorubicin emission in the range 520–550 nm corresponding to the emission filter in the CEM (Supplementary Figure 2b). As shown in Supplementary Figure 3a, the doxorubicin signal is relatively small compared to the total EGFP signal (also shown in Supplementary Figure 3b scaling approximately linearly as 1% of total detected photons per 1 μM doxorubicin). The observed decrease in intensity-weighted mean EGFP fluorescence lifetime with doxorubicin concentration (solid magenta line in Fig. 3e) cannot be explained by only the bleed-through of doxorubicin in this channel (solid red line in Fig. 3e), but is well-modeled by a combination of FRETing EGFP and doxorubicin emission (solid blue line in Fig. 3e).

To measure unknown concentrations of intracellular doxorubicin in vivo, we adopted an empirical approach. Briefly, we used the in vitro measurements discussed above to calibrate the expected change in the relative contributions of non-FRETing and FRETing H1-EGFP fractions with doxorubicin concentration

(see Fig. 3c, d, e and Supplementary Figure 3c, d and further description in Supplementary Note 1). The resulting calibration is represented by the blue line in Supplementary Figure 3c. While the EGFP fluorescence decay is well described by a single exponential decay, for chromatin tethered in a cell treated with doxorubicin we expect a distribution of FRETing and non-FRETing EGFP. As described in the Supplementary Note 1, the fluorescence from this distribution is approximated by a double exponential decay where the long component represents non-FRETing H1-EGFP and the short component represents the FRETing H1-EGFP. The long component was determined by fitting a single exponential decay model to the FLIM data obtained from untreated cells. The short component was then determined by globally fitting data from all (i.e. treated and untreated) cells to a double exponential decay whilst keeping the long lifetime fixed to the value determined from the untreated cells alone. Applying this approach to the multiphoton LSM EGFP decay data, which uses a shorter emission filter excluding any doxorubicin fluorescence, yielded EGFP$_{long}$ and EGFP$_{short}$ lifetime components of 2330 ps and 770 ps, respectively, (percentage non-FRETing H1-EGFPP contribution is shown by circles

and green line in Supplementary Figure 3c). The FRETing population fractions were then calculated as a function of doxorubicin concentration for both the multiphoton LSM and CEM data with the EGFP$_{long}$ and EGFP$_{short}$ lifetime components fixed at these values for the multiphoton LSM data and at 2460 ps and 770 ps for the CEM data. Supplementary Figure 3c shows the resulting plots of non-FRETing population fraction as a function of doxorubicin concentration. These interpolated curves serve as a look-up table to enable the FRETing population fraction to be estimated from a measured FRETing population fraction using the CEM (green and blue lines in Supplementary Figure 3c).

To demonstrate the consistency of this approach, we used the EGFP$_{long}$ and EGFP$_{short}$ fractions obtained from the in vitro LSM data, together with the expected contribution of bleed through from the doxorubicin emission, to simulate the expected decrease in intensity-weighted mean fluorescence lifetime that would be measured with the CEM device. Figure 3e and Supplementary Figure 3d shows the good agreement of this simulation with the actual CEM fluorescence lifetime data. The fraction of EGFP$_{short}$ increases most rapidly at low doxorubicin concentrations and starts to plateau at higher concentrations. The reduced sensitivity at higher concentrations may be due to saturation of DNA binding and demonstrates that the device is most sensitive in the hundreds of nM to low µM range.

**In vivo imaging of ovarian cancer metastases**. The data above indicate that the CEM can determine changes in EGFP fluorescence lifetime that are dependent on doxorubicin binding to DNA with sensitivity in the 100 nM range. To understand more about doxorubicin-DNA binding in vivo, we imaged a model of disseminated ovarian cancer in the peritoneal cavity. IGROV-1 cells expressing H1-GFP were injected into the peritoneal cavity of immuno-deficient mice. Within 10–14 days, tumor nodules were visible on the peritoneal wall, spleen, mesentery, and other organs. To perform intravital imaging, we anesthetized the mouse and made a small incision in the peritoneal wall to expose the tumor nodules. Forceps were used to gently manipulate the position of the tissue and a micro-manipulator was used to bring the CEM probe into contact with the tumor (Fig. 4a).

Images were then acquired with the CEM with excitation pulses at 488 nm and 80 MHz repetition rate with the commercial Cellvizio® scanner running at a frame rate of 8.5 Hz (Fig. 4b). This frame rate enabled multiple frames to be captured in between the major tissue movements caused by the breathing of the mouse; however, the trade-off was the small number of photons captured per frame. To circumvent this issue, we tracked the motion of the tumor tissue in $x$ and $y$, frame by frame, and compared the relative alignment of each frame to an initial reference frame using a normalized cross-correlation algorithm[25]. During large breathing-related tissue movement, the alignment metric regularly dropped below 0.9 (1 = perfect alignment); the frames were too different from the majority to be accurately aligned and were discarded (Fig. 4b–e—similar to the method of Krummel and colleagues[26]). A total of 85 frames were aligned and the photon arrival times for the aligned pixels were collated. This method typically enabled 400–1000 photons per pixel to be collected in 10 s. The improvement in the intensity decay curves as a function of imaging time is shown in Fig. 4f–h. These collated datasets were then thresholded and subject to erosion and dilation steps followed by a watershed algorithm to generate a mask of the nuclei (Fig. 4i–l). The fluorescence lifetime information for all the pixels in any given nucleus was then pooled and the lifetime analysis performed, yielding >10,000 photons per nuclei for analysis, resulting in a fluorescence lifetime for every nucleus in the field of view. These data demonstrate that the CEM can image ovarian metastases within the peritoneum with sufficient spatial resolution to unambiguously segment individual nuclei and acquire sufficient photons to fit to a double exponential decay model.

**Comparing delivery routes of doxorubicin for chemotherapy**. Having established the functionality of the CEM for in vivo FLIM imaging, we set out to image fluorescence lifetime changes following doxorubicin administration to mice. The delivery of chemotherapy to ovarian cancer patients is most frequently through an intravenous (IV) route, but in some countries the more direct route of intraperitoneal (IP) injection is used[27,28]. We delivered a fixed dose of 5 mg kg$^{-1}$ of doxorubicin via either the IP or IV route and imaged mice at either 1.5, 3, or 24 h. Multiple fields-of-view were imaged per mouse, leading to the acquisition of fluorescence lifetime data for >500 nuclei per mouse. 400–1200 photons per pixel were acquired (Fig. 5a). Attempts to image doxorubicin-treated tumors that did not express H1-EGFP were largely unsuccessful; only 30–120 photons per pixel could be acquired in the few fields-of-view where any signal was detectable (Supplementary Figure 4a) and this supports the predominance of EGFP-emitted photons in the signal that we acquire from H1-EGFP-expressing tumors. Figure 5b–h shows frequency distributions for the proportion of short component of the H1-EGFP lifetime, which equates to the H1-EGFP engaging in energy transfer with doxorubicin, and Fig. 5i–o shows the calculated doxorubicin concentration required to generate this shift in vitro. Intraperitoneal delivery of doxorubicin leads to large increases in the relative contribution of the short EGFP lifetime component. Approximately half the H1-EGFP has shifted to the short fluorescence decay component 1.5 h after doxorubicin injection. This indicates high levels of drug–target engagement equivalent to an average dose of ~20 µM in vitro; although, we note that mouse #3, (shown in Fig. 5c, j) did not reach such levels of drug–target binding. After 3 h, the frequency distributions of short H1-EGFP lifetime were across a broad range from ~0.1–0.6, (shown in Fig. 5e, l). There was significant heterogeneity between mice (each mouse is plotted in a different color in Fig. 5b–o) and, on top of this, there was also a considerable spread of short H1-EGFP relative contribution values in any single mouse (for example, mouse #1 and mouse #5 at 3 h following intraperitoneal delivery—shown in Fig. 5e, l). Overall, the range of relative contribution of the short decay component of H1-EGFP suggested a level of doxorubicin–chromatin binding equivalent to 5–40 µM or greater in all mice 1.5 h after intraperitoneal injection. This was reduced 3 h postinjection, but the majority of cells in mice (4 out of a total of 5 mice, shown in Fig. 5e, l) still had doxorubicin binding equivalent to ~5 µM or more. After 24 h (shown in Fig. 5g, n for IP), the H1-EGFP short fluorescence lifetime distributions were centered between 0.1 and 0.2, approximately equivalent to 1 µM doxorubicin in vitro.

A key motivation for developing a confocal endomicroscope capable of FLIM was to enable repeated imaging of the same tumor nodules with minimal tissue disruption. We therefore sought to perform repeat imaging on the same nodule at different times following delivery of doxorubicin. Supplementary Figure 5 shows fluorescence lifetime measurements of three different nodules at 45, 90, and 180 min after intra-peritoneal injection of 5 mg kg$^{-1}$ doxorubicin. Between image acquisitions, the peritoneal cavity was kept closed to avoid dehydration. Consistent with the observations across different mice (Supplementary Figure 5 - subplot sets (a–c), (d–f), and (g–i) for mouse #1, #2 and #3, respectively), doxorubicin–chromatin engagement declined between 90 and 180 min. These data demonstrate the capability for longitudinal imaging of tumors.

Comparison of intravenous and intraperitoneal delivery revealed the striking inefficiency of the intravenous route (Fig. 5). The majority of the nuclei in the frequency distribution plots had <0.2 of short fluorescence lifetime component, equivalent to 1 μM or lower equivalent doxorubicin concentrations in vitro.

Occasional patches of cells with greater drug–target engagement were observed (see mouse #3 at 1.5 h and mouse #2 & #4 at 3 h in Supplementary Figures 6 and 7). These data indicate that intravenous delivery of doxorubicin is sub-optimal compared to intraperitoneal delivery.

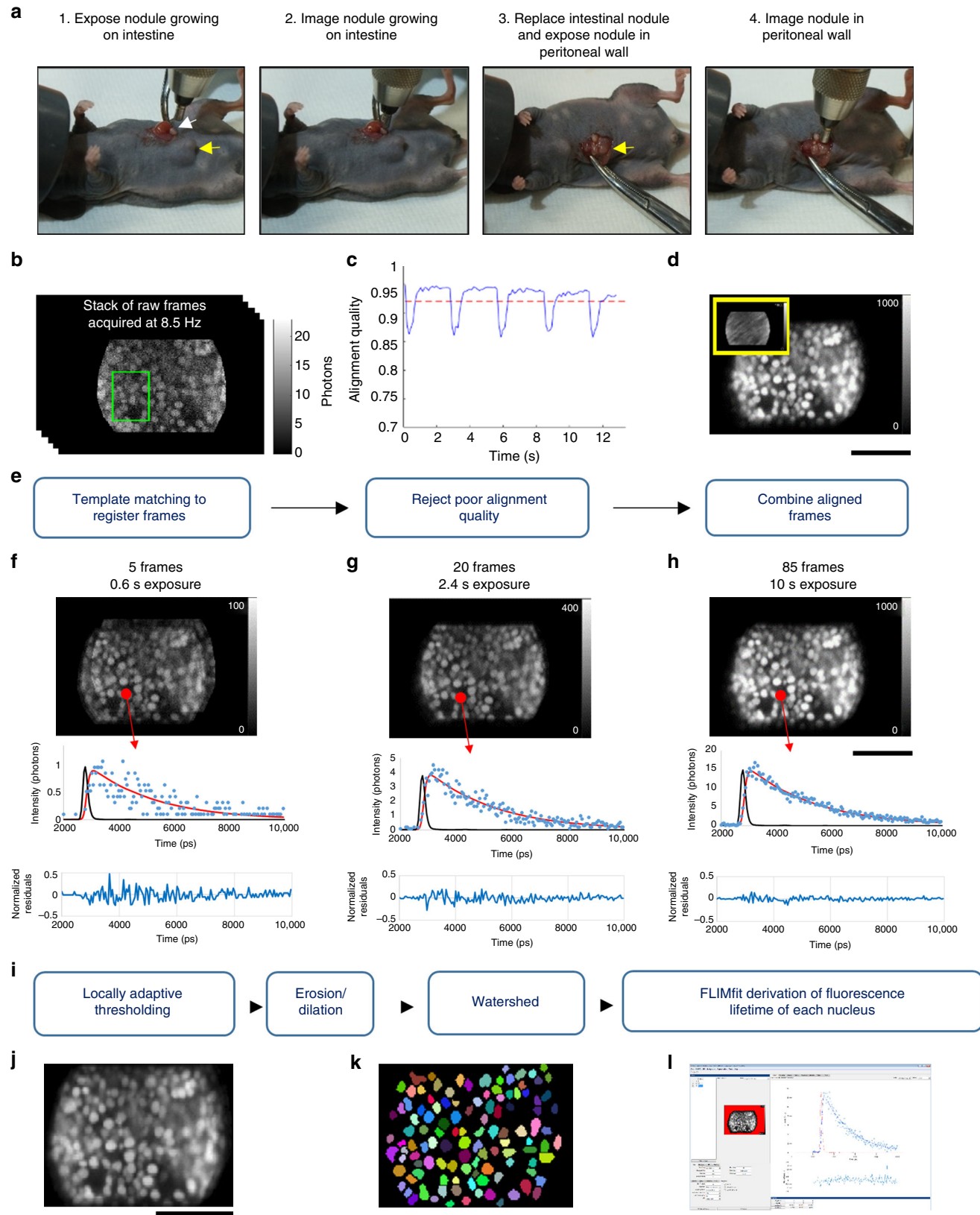

**Intra-tumor heterogeneity**. The distributions shown in Fig. 5 indicate significant inter-mouse heterogeneity following IP delivery of doxorubicin, but do not address heterogeneity within a single mouse. This is a particularly critical question as intra-tumor heterogeneity is linked to poor patient outcomes[29,30]. We first compared different nodules in the same mouse: two on the spleen and one on the peritoneal wall (Fig. 6a–i). Considerable heterogeneity was observed: for example, the spleen tumor nodule designated T2 showed more doxorubicin–chromatin binding than the peritoneal nodule designated T1. Intriguingly, one region of spleen tumor nodule T3 showed almost no doxorubicin binding. Thus, within the same mouse at the same time, one region of an ovarian metastasis has doxorubicin–chromatin binding equivalent to an in vitro dose of 20 μM (Fig. 6g – T2, view #2), while another region of a different metastasis has <1 μM equivalent dose (Fig. 6i – T3, view #3). The images in Fig. 6 also indicate that there can be large variation between fields-of-view of the same tumor nodule, and even within a field of view. Figure 6j–l present analysis of six fields-of-view of a single nodule in a mouse 3 h after IP delivery of doxorubicin. Even within this limited region, there is variation in doxorubicin–chromatin binding equivalent to a 2–40 μM dose range in vitro (Fig. 6m–o show analysis for six fields-of-view in a control mouse with no treatment). A comprehensive presentation of the heterogeneity between the different fields-of-view in every doxorubicin-treated mouse is shown in Supplementary Figures 6 and 7. This further reinforces the high variability in drug–target engagement that is a feature of many tumors, even in an experimental context supposedly optimized for consistency.

**Discrepancy between doxorubicin effect in vitro and in vivo**. The quantitative nature of fluorescence lifetime measurements means that we are reliably able to compare the level of doxorubicin–chromatin engagement in vitro and in vivo. This then allows us to investigate if the same level of drug–target engagement has the same consequences in vitro and in vivo. Figure 5 shows that IP delivery of doxorubicin leads to a distribution of doxorubicin binding, but notably with almost every cell experiencing a concentration equivalent to >1 μM in vitro. To determine the effect of doxorubicin on IGROV-1 cells in vitro, we treated them with the same range of concentrations used to probe changes in fluorescence lifetime in vivo and measured their production of luciferase from a stably integrated transgene after 48 h. This metric gives an indication of the number of viable cells and has the key advantage that it can also be used in vivo enabling direct comparison with in vitro analysis. Figure 7a shows that 0.9 μM doxorubicin reduced cell viability by almost 60% after 48 h. This predicts that IP delivery of 5 mg kg$^{-1}$ of doxorubicin should lead to a similar reduction in luciferase signal in vivo. To determine if doxorubicin did indeed have such efficacy in vivo, we injected IGROV-1 cells stably expressing luciferase into the peritoneal cavity of immune-deficient mice. After 16 days, the mice were split into two groups and one injected with PBS as a control and the other with 5 mg kg$^{-1}$ doxorubicin. Luciferase imaging was performed before injection and after 48 h. Figure 7b–g shows that control mice had similar tumor burden before and after PBS injection. Strikingly, doxorubicin-treated mice also had similar tumor burden two days after doxorubicin treatment. This indicates that doxorubicin is dramatically less effective in vivo than in vitro despite efficient target engagement after 3 h.

**Discussion**
There are many reasons why cancer therapies fail, including: sub-optimal drug access, expression of multidrug exporters, compensating signals from the tumor microenvironment, and genetic resistance mechanisms[1–4]. A greater understanding of the relative contributions of these mechanisms should improve cancer outcomes. In this work, we describe the modification and utilization of a confocal fiber-optic endomicroscope for obtaining high resolution fluorescence lifetime data in vivo. By exploiting the spectral properties of doxorubicin to modulate the fluorescence lifetime of GFP tethered to chromatin via histones, we have been able to map drug–target engagement with cellular resolution. The independence of fluorescence lifetime measurements on signal intensity enable the direct comparison of doxorubicin–chromatin binding between in vitro and in vivo experiments, and between different in vivo delivery routes. Our FLIM endomicroscope measurements revealed large variations in doxorubicin–target binding between tumors, between different tumor nodules, and even within a single field of view. The reasons for this heterogeneity are unclear. Inter-tumor and inter-nodule variation may reflect localized differences in the metabolic state and rate of drug uptake. Dense extracellular matrix may also act as a local barrier to drug access. Alternatively, differences in interstitial tumor tissue pressure may play a role[31]. Very local heterogeneity, as observed in Fig. 6, may be caused by variation in the expression of drug efflux pumps; indeed, several studies have suggested that cancer stem cells have elevated levels of this class of molecules[32]. The overall theme of localized drug response is consistent with previous intravital imaging studies[33–36]. Nakasone et al. directly imaged doxorubicin fluorescence and attributed inter- and intra-tumor heterogeneity to differing activity of myeloid cells affecting vascular permeability[37]. Hirata et al[4] observed reduced efficacy of BRAF inhibitors in areas with high tumor stroma: in this study, tumor-stroma signaling was responsible for the reduced drug efficacy, not defects in drug access.

**Fig. 4** Intravital image acquisition and analysis. **a** Images show examples of two nodules being imaged. The white arrow indicates a nodule attached to the intestine and the yellow arrow indicates a tumor mass growing into the peritoneal wall. **b–e** Schematic of analysis workflow for in vivo CEM FLIM of IGROV-1 H1-GFP labeled tumors in mouse model. Raw integrated intensity images produced by CEM at 8.5 Hz (**b**) are aligned using template matching (example template shown with green box in **b**) based on normalized cross-correlation. The alignment quality is periodically degraded by breathing (example shown in **c**) and frames that are severely affected by motion are rejected; the remaining frames are combined following alignment to generate high signal-to-noise image data (as shown in **d**) that are suitable for FLIM analysis. The inset image with a yellow border in **d** shows the same 10 second acquisition but without applying the motion compensating alignment algorithm. **f–h** show how increasing the number of frames that are combined increases the accuracy of fitting to a single exponential decay model. As frame accumulations are increased from 5 to 85 frames, FIFO FLIM integration times increase from sub-second to 10 s and the number of photons per pixel increases. This is associated with a decrease in the residuals following fitting of the FLIM data to the single exponential decay model. **i** image segmentation was used to calculate lifetimes on a nucleus by nucleus basis: a locally adaptive thresholding routine was used to form an initial segmentation guess and then erosion and dilation steps were used to separate contiguous nuclei, followed by application of the watershed.m function in MATLAB. The segmentation masks were then loaded into FLIMfit along with the raw TCSPC FIFO data. Integrated intensity image from aligned raw TCSPC FIFO data, watershed segmentation binary image, and FLIMfit graphical user interface are shown in **j**, **k**, and **l**, respectively

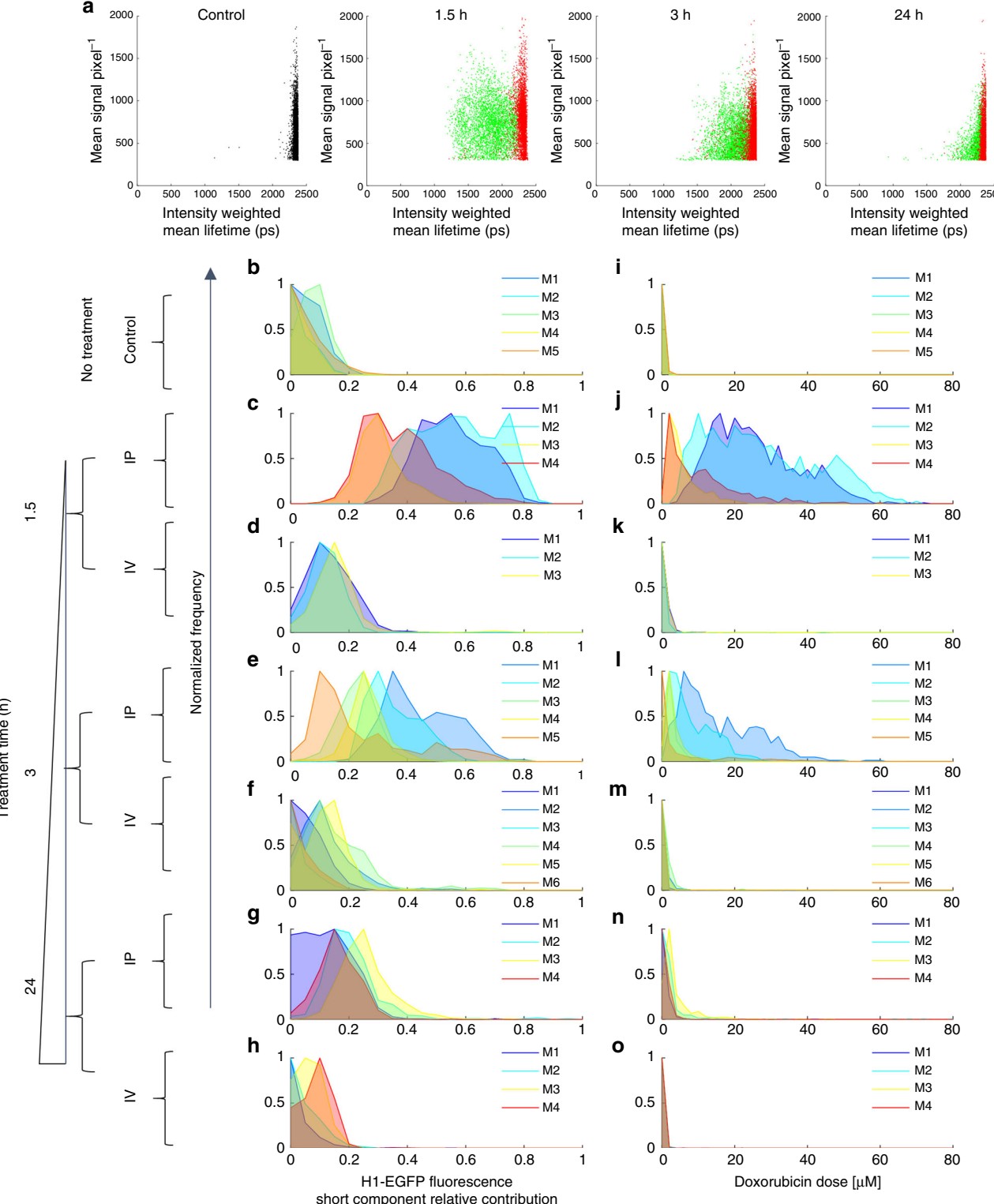

**Fig. 5** Comparison of intravenous and intra-peritoneal delivery routes. CEM FLIM data from murine cancer model based on IGROV-1 cells injected intraperitoneally and allowed to grow into tumors over 14–21 days. Subsequently mice were treated with 5 mg kg$^{-1}$ doses of doxorubicin by intraperitoneal (IP) or intravenous (IV) injection. At 1.5, 3, and 24 h post dosing, repeat models underwent minor surgery to expose tumors for imaging with the CEM. **a** scatter plots of intensity-weighted mean H1-EGFP fluorescence lifetime against mean intensity for each for each cell nucleus from control mice (black), mice that underwent IP delivery of drug (green), and mice that underwent IV delivery of drug (red). The drug incubation times 1.5, 3, and 24 h are shown above each plot in **a** for doxorubicin-treated mice. **b–h** Histograms show the distribution of calculated population fraction of FRETing H1-EGFP with higher values indicating increasing doxorubicin binding to chromatin; **i–o** Histograms show intracellular doxorubicin concentration estimated according to in vitro calibration as discussed in Supplementary Note 1. Individual mice (M) are represented in each histogram by different colors. Control mice distributions are shown in **b**, **i** and treated mice distributions are shown in **c–h**, **j–o**. Biological replicates for each condition are as follows: $n = 5$ control mice; $n = 4$ IP mice at 1.5 h; $n = 3$ mice at IV 1.5 h; $n = 5$ mice at IP 3 h; $n = 6$ mice at IV 3 h; $n = 4$ mice at IP 24 h; $n = 3$ mice at IV 24 h

A key feature of using fluorescence lifetime to read out doxorubicin–chromatin interactions is the ability to relate in vitro and in vivo levels of drug–target engagement. This reveals strikingly different fates of cells that experience equivalent levels of doxorubicin–chromatin binding. In vitro concentrations over 1 µM are sufficient to kill IGROV-1 cells; however, in vivo there is very little reduction in tumor burden even though doxorubicin–chromatin interactions are equivalent to ~10 µM in vitro dosing. This indicates that there is substantial support for IGROV-1 cells from the tumor microenvironment that is capable of enabling viability even in cells with disrupted topoisomerase II—the main molecular target of doxorubicin–DNA complexes.

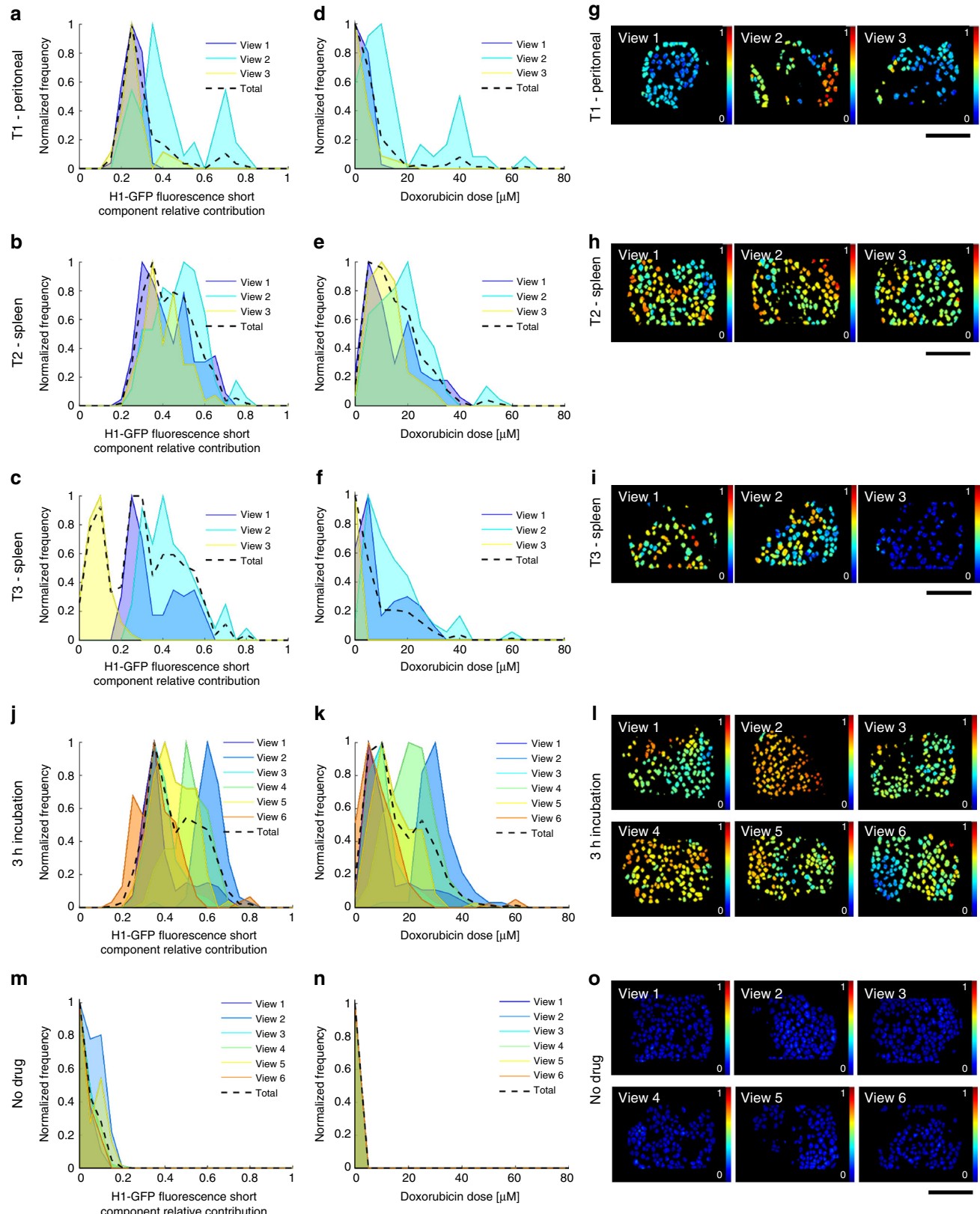

Several studies have shown that factors produced by the tumor stroma can ameliorate the effect of genotoxic therapies[38,39]. This is likely the reason for part of the discrepancy between in vitro and in vivo cell killing. Another possibility is that the slower rates of proliferation in vivo allow more time for topological problems caused by reduced topoisomerase II function to be resolved.

The quantitative nature of fluorescence lifetime measurements also enables direct comparison of differences between drug delivery routes. This revealed that intravenous delivery was significantly less effective than peritoneal injection. A partial explanation may be that our analysis concentrated on the cells near the outer surface of tumor nodules that may be further from blood vessels; however, we think this is unlikely as H&E staining did not reveal differences in cell viability close to the tumor edge (Supplementary Figure 7). Intravenous delivery is the mainstay of chemotherapy delivery for ovarian cancer patients. The data presented here suggest that a re-evaluation of delivery route is warranted.

Like several cancer types, ovarian cancer detection relies on a combination of nuclear, laparoscopic/endoscopic, and histological imaging techniques. The device described here has the sensitivity and small dimensions to be deployed for clinical applications; we note that in a clinical setting confocal endomicroscope/laparoscope devices can be passed through the working channel of larger endoscopes. In the long term, future iterations of the device described here combined with appropriate contrast agents could provide rapid intra-operative evaluation of histology, leading to beneficial reductions in the length of operations and even more accurate disease resection.

To conclude, in this work we have imaged doxorubicin binding to chromatin to map drug–target engagement with unprecedented resolution and accuracy in vivo. As well as providing information on the relative efficacy of different drug delivery strategies, this approach has revealed striking heterogeneity in drug–target engagement between different tumor nodules, and even within single fields-of-view. Further, we document strikingly different outcomes to the same level of doxorubicin binding to chromatin in vitro and in vivo. These observations highlight the challenges in improving chemotherapy delivery and efficacy.

## Methods

**Cell lines**. IGROV-1 cell lines were cultured in $CO_2$ dependent media with 10% fetal bovine serum and 1% Pen Strep at 37 °C. Before experiments, cells were grown to 80% confluence. For measuring doxorubicin uptake by fluorescence an IGROV-1 cell line stably expressing GFP fused to Histone-1 (H1) was made using the PiggyBac transposon system. As a control to show that effect of doxorubicin on GFP depends on whether it is fused to H1 or not, a stable whole cell expression of GFP by lentiviral transfection and selection by Geneticin was made. For bioluminescence imaging of xenograft tumors, all IGROV-1 cell lines were made to stably express firefly luciferase.

To investigate the effect of doxorubicin on other histones, IGROV-1 cells were transiently transfected with a Histone-2B-GFP plasmid (gift from Kurt Anderson) using the Lipofectamine® 2000 reagent.

IGROV-1 cells were obtained from Crick institute cell services and confirmed as IGROV-1 by Short Tandem Repeats (STR) profiling and no mycoplasma was detected.

**In vitro experiments**. IGROV-1 cells were grown to 80% confluence in 75 ml flasks before being re-plated in 12- or 24-well plates or 35 ml glass bottomed dishes and allowed to attach to the surface for 24 h before experiments.

To study how the fluorescence of GFP labeled H1 labeled IGROV-1 cells changes with doxorubicin treatment, fluorescence intensity and lifetime distributions were measured from cells after 3 h of incubation with doxorubicin of varying concentrations (0, 0.18, 0.9, 1.8, 9, 18 μM) by serial dilutions of a stock solution with PBS. After 3 of h, cells were washed in PBS then fixed for 20 min in 4% PFA. Cells were then imaged in PBS. Doxorubicin hydrochloride (Sigma-Aldrich, D1515-10 mg) was dissolved in PBS to a concentration of 9 mM and stored at −20 °C.

**In vivo experiments**. Murine xenografts were prepared by intraperitoneal (IP) injection of IGROV−1 cancer cells. IGROV-1 cells were grown to 80% confluence before being trypsinized and re-suspended in PBS at a concentration of $2 \times 10^7$ cells per ml. $2 \times 10^6$ cells were injected into ICRF nude mice. After 14 days postinjection, the presence of intraperitoneal tumors was confirmed by bioluminescence imaging. Briefly, an IVIS bioluminescence imaging system was used to image isoflurane anesthetized mice. 100 μl of D-luciferin (luciferase substrate) at 30 mg ml$^{-1}$ was injected IP 10 min before recording of bioluminescence images. The presence of peritoneal tumors was confirmed if bioluminescence signals from the peritoneum were above background noise 10–30 min after D-luciferin injections. Following confirmation of tumors, in vivo fluorescence imaging experiments were carried out after 21 days. To study differences in drug uptake between intravenous or intra-peritoneal delivery, prior to imaging mice were subject to IP or IV doxorubicin-based chemotherapy for 1.5, 3, or 24 h. Imaging involved terminal procedures, mice were anesthetized then peritoneal tumors were exposed by minor surgery and inspected with the CEM.

All animal model procedures were approved by The Francis Crick Institute Biological Ethics Committee and UK Home Office authority provided by Project License 70/8380.

**Spectrally resolved confocal microscopy**. For spectrally resolved fluorescence imaging, a Zeiss 780 microscope equipped with a 32-Channel GaAsP spectral detector (QUASAR, detection unit, Carl Zeiss) was used with single photon excitation at 488 nm. For all experiments, a 63× water immersion objective was used (C-Apochromat 63× /1.2 W Corr., part # 441777-9970-000). Autofluorescence, dish background fluorescence and room noise were estimated from measurements of wells containing unlabeled cells with no treatment and subtracted from data accordingly.

**FLIM with multiphoton microscopy**. For multiphoton FLIM microscopy, a Simple Tau 150 TCSPC system (B&H) comprised of two hybrid GaAsP photomultipliers connected to two SPC-150 TCSPC cards was integrated into the LSM system (Carl Zeiss, LSM780). To be compatible with multiphoton laser-scanning microscopy, the detectors were placed in the non-descanned detection path and a 690 nm dichroic was placed in the microscope's filter cube turret. To block any residual near infrared (NIR) laser light, a 690 nm short pass emission filter was placed in the non-descanned detection path before the photomultipliers. For fluorescence emission a 465–495 nm bandpass filter was used before one of the photomultipliers.

For TCSPC imaging, B&H SPCM acquisition software was used in first-in-first-out (FIFO) mode where photon arrival times were measured relative to pixel, line, and frame clocks from the Zeiss LSM780 microscope and excitation pulses from the laser. Images of 256 × 256 pixels and with 256 time bins were used across all experiments. To avoid lifetime artefacts due to pulse pile-up, ADC count rates were kept at or below ~1% of the laser pulse repetition rate (90 MHz). To collect ~1000 photons per pixel, exposure times ranging from 60 to 120 s were required. To measure the instrument response function (IRF) gold nanorods were used to generate ultra-fast broadband luminescence[40].

**FLIM with confocal endomicroscopy**. A commercial CEM (Mauna Kea technologies, Cellvizio®) was adapted for TCSPC[14]. A mode-locked frequency-doubled Ti:Sapphire laser (Spectra-Physics, BB MaiTai) was coupled to the commercial scanning Cellvizio® unit using a single-mode optical fiber (Thorlabs, HP-405). The

**Fig. 6** Intra-nodule and regional heterogeneity in doxorubicin chromatin binding. CEM FLIM data from murine cancer model based on IGROV-1 cells with mice treated with 5 mg kg$^{-1}$ doxorubicin doses by IP injection. **a**, **b**, **c** show intra-nodule variation in normalized histograms of H1-GFP FRETing population fractions 3 h after IP delivery of drug, together with histograms of estimated intracellular doxorubicin concentration, **d**, **e**, **f** for three fields-of-view of a tumor nodule (T1) on the peritoneal wall, a tumor nodule (T2) on the spleen, and a second tumor nodule (T3) on the spleen in the same mouse, respectively. **g**, **h**, **i** show the corresponding maps of the H1-EGFP FRETing population fraction (where color scale from blue to red spans 0–1). **j** shows normalized histograms for H1-GFP FRETing population fractions and **k** shows estimated intracellular doxorubicin concentration for 6 fields-of-view from a single nodule in a control mouse and **l** shows the corresponding map of the H1-EGFP FRETing population fractions. **m** shows normalized histograms for H1-GFP FRETing population fractions and **n** shows estimated intracellular doxorubicin concentration for 6 fields-of-view from a single nodule in a mouse, 3 h after intra-peritoneal delivery of 5 mg kg$^{-1}$ doxorubicin and **o** shows the corresponding map of the H1-EGFP FRETing population fraction. For all images the scale bar is 120 μm

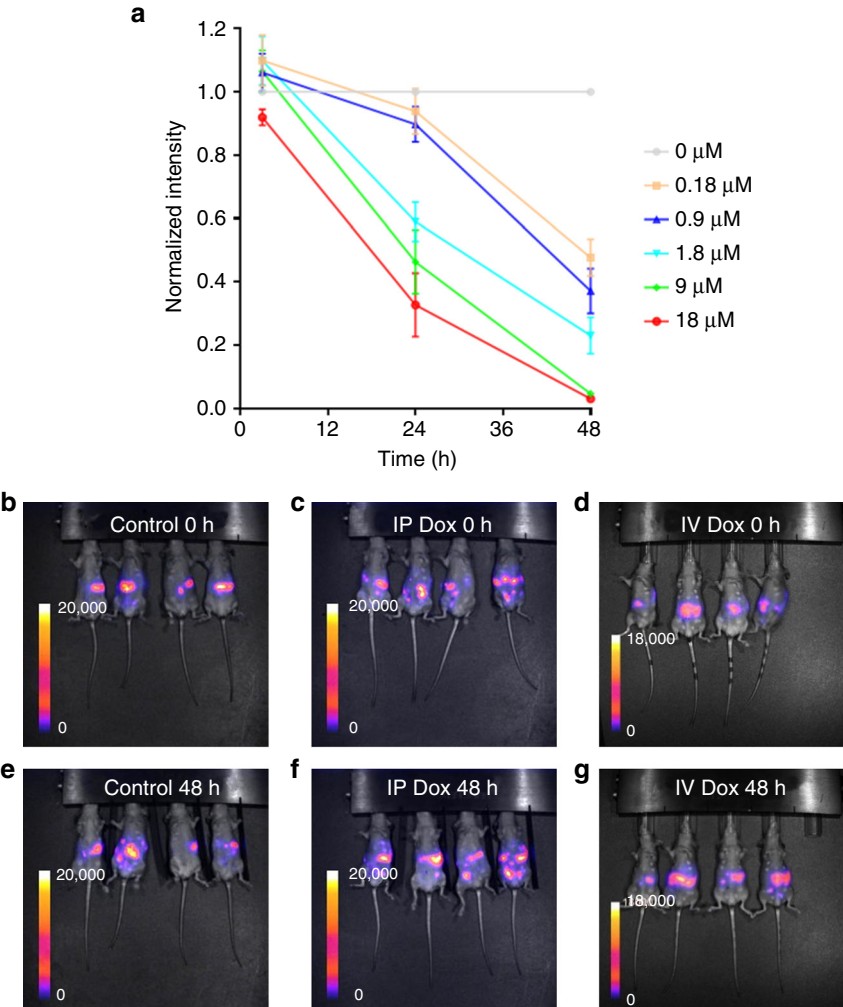

**Fig. 7** Differential doxorubicin efficacy in vivo and in vitro. **a** Chart shows effect of 0, 0.18, 0.9, 1.8, 9, and 18 μM doxorubicin treatment on the viability of IGROV-1 stably expressing luciferase in vitro. Viability was assessed by luciferase activity in 24-well plate ($1 \times 10^5$ cells well$^{-1}$). Each data point represents the average intensity relative to that for 0 μM plotted at time = 0 h. The error bars represent the s.d. for $n = 3$ biological replicates i.e., three 24-well plates. **b–g** images show the effect of doxorubicin on IGROV-1 tumor burden in vivo. **b–d** show luciferase imaging of mice immediately (0 h) following control (**b**), intraperitoneal (**c**) or intravenous (**d**) injection with control PBS or 5 mg kg$^{-1}$ doxorubicin. **e–g** show the corresponding sets of mice 48 h postinjection. Biological replicates of $n = 4$ mice were used for each luciferase imaging condition

fiber aperture acted as a pinhole for excitation light in a confocal arrangement with a photomultiplier. A dichroic beamsplitter reflected laser light to the commercial endoscope optical scanning unit and transmitted returning fluorescence to the photomultiplier (PMT, PMH 100-1, B&H). In the commercial scanning unit, a 4 kHz resonance mirror provided the fast line scan direction and a galvanometric mirror operating at 8.5 Hz provided the slow scan direction (i.e., a frame rate of 8.5 Hz). The focused laser spot is raster-scanned across the distal end of the CEM's coherent fiber bundle.

A prototype Cellvizio® Mini O probe designed for minimal autofluorescence background was used[41]. The coherent fiber-optic bundle consists of ~30,000 cores with a core-to-core spacing of ~3 μm and core diameters of ~2 μm. A mini objective at the probe's distal end results in a final (manufacturer specified) lateral and axial resolution of 1.4 μm and 10 μm, respectively, and a field of view of 240 μm and working distance of 60 μm when used in the intended configuration (i.e., using the internal laser light and detector). The diameter of the distal tip is 2.6 mm.

For FLIM, the PMT signals and line and frame clocks from the scanning unit electronics and laser were fed to a SPC-830 TCSPC card (B&H) to register detected fluorescent photons' arrival times and spatial positions on the fiber-optic bundle end face. One computer controlled the commercial scanning system, and another controlled the FLIM system including the laser and TCSPC electronics.

For live FLIM updates, FIFO acquisition mode was used and in-house written software, including a graphical user interface (GUI), was used to compile photon events into an image at an update rate of 8.5 Hz. For each measurement, at least 30 s of FLIM data was recorded, with a subset of 85 well-aligned frames (corresponding to 10 s of FIFO TCSPC data) being used for subsequent FLIM

analysis, as depicted in Fig. 4. To align the frames of each dataset, a sub-region of the first frame was selected to serve as a template for alignment of the following frames relative to this first reference frame. The lateral shift between frames was determined from the location of the peak of the normalized image cross-correlation (calculated using the OpenCV C++ template-matching function matchTemplateOCV.cpp available in MATLAB). Typically, frames with a value above 0.9 were retained for subsequent FLIM analysis, but in some cases (usually those with lower signal-to-noise ratio) a cut-off of down to 0.8 was used. Frames having a low normalized cross-correlation, e.g., those significantly distorted due to breathing, did not contribute to the FLIM analysis and the image distortion over each sequence of the 85 selected frames was observed to be much less than the inter-nuclear distance. Thus, the risk of erroneously mixing photon arrival times between nuclei in the calculation of the mean lifetimes of each cell nucleus was low.

To indicate lifetime variation across the field of view in real time, intensity-weighted FLIM images based on mean photon arrival times were displayed by the GUI. Data recording was controlled by buttons on the GUI. To avoid lifetime artefacts due to pulse pile-up, ADC count rates were kept at or below ~1% of the laser repetition rate (80 MHz). For analysis, FIFO datasets were converted to tiff stacks with image dimensions of 118 × 172 pixels.

The Ti:Sapphire laser was tuned to 976 nm and frequency-doubled to 488 nm, and a 505 nm dichroic was used to direct laser light to the commercial CEM laser-scanning unit and fluorescence to the photomultiplier[41]. A bandpass emission filter with transmission in the spectral range 520–550 nm was placed in front of the PMT. Per FLIM image, acquisition times were up to 60 s in order to provide 10 s (or 85 frames) of useable data following gating and image registration so as to end up with ~1000 photons per pixel.

To measure the IRF, scattered laser light was detected at the photomultiplier by placing a rough surface in the beam path before the commercial scanning unit. To avoid damage to the detector, the bandpass fluorescence emission filter was replaced with a neutral density filter to attenuate excitation light by a factor of more than 1000 and the laser power was reduced to ~1 mW. Because of the difference in optical path length between the scattered light and fluorescence from the sample due to material dispersion in the fiber, it was necessary to image a reference dye to obtain a calibration of the excitation pulse arrival time across the image. The fluorescence decay data measured with the CEM was then used to determine the temporal offset between the system IRF and the fluorescence decay profile. For each day of measurements, this offset was determined and used for subsequent analysis.

FLIM datasets were first corrected for background fluorescence, e.g. excited in the fiber-optic bundle, by subtracting a value measured when the endomicroscope tip was placed in water. Because this background is significant, a first order correction to background based on TCSPC counting statistics was used (see ref.[42]).

The GUI and the script to convert the binary files to FLIM images are available on request.

**Summary of spectral channels used across instruments**. The excitation wavelengths and emission detection bands for the two instrument setups used, namely the Zeiss LSM multiphoton FLIM and CEM FLIM, are summarized in Table 1. The detection band used with the multiphoton LSM was used to measure the fluorescence intensity and lifetime of H1-EGFP. The detection band used with the CEM FLIM configuration was used to measure fluorescence intensity and lifetime of H1-EGFP and fluorescence intensity and lifetime of doxorubicin.

**Image analysis**. To extract mean nuclear intensities and lifetimes, a nuclear segmentation routine was implemented. Following subtraction of background, data was smoothed to reduce noise then a locally adaptive thresholding algorithm was used to locate nuclei. An adaptive image thresholding algorithm using minimax optimization[43] was used. This thresholding method avoided manually setting thresholds for each image and handled spatial variations in sample brightness within a field of view. To remove irregular shapes and separate touching objects morphological operations of erosion and dilation were implemented using MATLAB's imerode.m and imdilate.m functions with a square (three pixels edge size) and disc (two pixels radius) kernel, respectively, to split touching nuclei. Objects with total areas less than 20 pixels were rejected using MATLAB's bwareaopen.m function and the remaining regions were used as inputs for MATLAB's watershed.m function. For marker-controlled watershed, we made use of the prior knowledge that nuclei regions will typically be circular in shape so the distance transform of the smoothed, eroded and dilated binary images should produce one maxima per nuclei. By inverting the distance transform, each minima forms a catchment basin for the watershed algorithm.

**FLIM analysis**. FLIMfit was used for the analysis of all datasets (http://flimfit.org). Decay analysis was performed for pixels with an integrated photon count of >300. $3 \times 3$ spatial averaging was used to reduce noise. Segmentation masks were generated by the method described in the section above. For in vitro datasets, two FLIM analyses were used. In the first instance, lifetime contrast was shown by fitting on a pixel-wise basis across each segmented region (a cell nuclei) to a single exponential decay model. In the second instance, FLIM data was fitted to a double exponential decay model. The two decay components represented the FRETing and non-FRETing sub-populations of H1-EGFP. The non-FRETing lifetime was determined by globally fitting FLIM data from control experiments (i.e., cells or mice without doxorubicin treatment) to a single exponential decay model. For in vitro measurements, this consisted of three fields-of-view containing a total of 100 or more cell nuclei. For in vivo measurements, this consisted of three or more fields-of-view and typically more than 500 nuclei per mouse. The FRETing lifetime was determined by globally fitting the in vitro LSM FLIM data from cells treated with 0, 0.18, 0.9, 1.8, 9, 18 μM doxorubicin for 3 h to a double exponential decay model with the long component fixed to the non-FRETing lifetime determined from the LSM FLIM control experiments.

These FRETing and non-FRETing lifetimes were then used to fix the long and short lifetime components in a double exponential decay model while keeping their relative contribution as a free fitting parameter. This model was then used to find relative contributions of FRETing and non-FRETing contributions across pixels in each cell nuclei segment for in vitro date sets with known treatment doses and for in vivo datasets with unknown treatment doses. For each cell nuclei segment, a pixel-wise average of both the intensity-weighted mean lifetime and the relative contributions of the long and short lifetime components was calculated. All cell nuclei values were combined into a grouped dataset, with each linked to the particular mouse, treatment delivery method and treatment time.

**Statistical methods**. For all experiments, samples sizes are reported in figure legends. No estimates of required sample sizes were calculated prior to in vitro or in vivo experiments. None of the samples or animals were excluded from the experiment. For in vivo experiments, no randomization or blinding was performed.

**Table 1 Excitation and emission filters used with multiphoton LSM FLIM and CEM FLIM**

| Instrument | Excitation wavelength (nm) | Emission filter (nm) |
|---|---|---|
| Multiphoton LSM FLIM | 900 | 465–495 |
| CEM FLIM | 488 | 520–550 |

In Fig. 2e, the difference in fluorescence lifetime between cells treated with DOX and untreated controls was considered for each construct using an unpaired, unequal variances two-tailed $t$-test in Microscoft Excel. First, following segmentation into separate cell nuclei regions, FLIM data was analyzed by fitting a single exponential decay on a pixel-wise basis using FLIMfit. For the two-tailed $t$-test, mean single exponential fitted lifetimes per cell nuclei segment were used and fitted fluorescence lifetime distributions were treated as normally distributed. At 0.1% significance level and quoting to 1.s.f, for free EGFP, H1-EGFP, and H2-EGFP the $P$-values were 0.4, $1 \times 10^{-7}$, and $5 \times 10^{-4}$, respectively, and the degrees of freedom were 14, 12 and 6, respectively.

**Code availability**. MATLAB code for generating aligned FIFO images and subsequently accumulated FIFO FLIM data is available under an open source license at https://doi.org/10.5281/zenodo.1249004.

**Data availability**. The raw FIFO data generating aligned FIFO images and subsequently accumulated FIFO FLIM data is available under an open source license at:
In vivo data: https://doi.org/10.5281/zenodo.1249023
In vitro data: https://doi.org/10.5281/zenodo.1249014
The aligned FIFO FLIM image data from this work is available under an open source license from Imperial College London's OMERO server at:
In vivo data: https://omero.bioinformatics.ic.ac.uk/omero/webclient/?show−project−4755
In vitro data: https://omero.bioinformatics.ic.ac.uk/omero/webclient/?show=project-4754

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

## Acknowledgements

We thank members of both Imperial Photonics group and the Sahai lab for advice and critical comments on this study. In particular, we would like to thank Eishu Hirata for assistance with intravital imaging and Sunil Kumar for assistance with the confocal endomicroscope FLIM. H.S., H.K., S.H., and E.S. were funded by the Francis Crick Institute, which receives its core funding from Cancer Research UK (FC001144), the UK Medical Research Council (FC001144), and the Wellcome Trust (FC001144). P.F., C.D., I.M., G.K., and H.S. were funded by UK Engineering and Physical Sciences Research Council (EPSRC) grant EP/F040202. HS acknowledges a Ph.D. studentship funded by EPSRC.

## Author contributions

Contributing authors: H.S., H.K., S.H., I.M., G.K., C.D., P.F., and E.S.. Conceptualization: H.S., E.S., C.D., and P.F.; Investigation: H.S., S.H., H.K., G.K., I.M., C.D., E.S., and P.F.; Methodology: H.S., S.H., C.D., P.F., and E.S.; Software: I.M., H.S., G.K., C.D., and P.F.; Analysis: H.S., C.D, E.S., and P.F.; Writing: E.S, H.S, C.D., and P.F.; Writing—review & editing: E.S, H.S, C.D., and P.F.; and Resources: C.D., P.F., and E.S..

## Additional information

**Competing interests:** The authors declare no competing interests.

