## [Peer Review File · Nature Communications]

Reviewers' comments:

Reviewer #1 (Remarks to the Author):

This is a very comprehensive and nicely written manuscript addressing an important problem in cancer therapy, i.e. the efficacy of drug delivery and drug-target engagement. The authors present a systematic approach to imaging (both in vitro and in vivo) of doxorubicin binding to chromatin using fluorescence lifetime imaging confocal endomicroscopy. The imaging results are significant while intriguing.

A few clarifications are needed in order to better evaluate the ability of FLIM microendoscopy to quantify the drug-target engagement.

1. A 80 MHz rep rate laser was used in the study. At such high rep rate the sample/fluorophores can be photobleached. This aspect needs to be addressed as it could affect the validity of the results.

2. A method that allows for tracking the motion of the tumor in vivo was used to account for the small number of photons captured per frame. The shortcomings of this method need to be discussed as they might affect the results from the in vivo measurements. This could contribute to the perceived/reported heterogeneity in doxorubicin chromatin binding.

3. The cellular metabolism in vivo (animals) differs from the in vitro (cell culture). Does that affect the doxorubicin-chromatin engagement and potentially contribute to the discrepancies observed between doxorubicin effects in vitro vs in vivo?

Reviewer #2 (Remarks to the Author):

In their manuscript, Sparks and colleagues describe the development of a novel quantitative microscopy approach based on FLIM using a confocal endomicroscope to visualize and quantify drug-target engagement in vivo. In the first part of the manuscript the technique is tested and validated. In the second part the authors show that the confocal endomicroscopy technique can be used to visualize the binding of doxorubicin to chromatin in a model of metastatic ovarian cancer. In this model, they reveal a large intra- and inter-tumor heterogeneity in chromatin binding by doxorubicin. Moreover, the data shows that the mode of drug administration, i.e. intra-peritoneal (IP) or intra-venous (IV), leads to a striking variation in drug delivery in this tumor model.

The development of the quantitative confocal endomicroscopy is an impressive effort and the presented biological findings are very interesting. Nevertheless, the authors need to address my major points before I can recommend to publish this manuscript at Nat Commun.

Major comments:

- In the first part of the manuscript the authors describe how they have used FLIM endomicroscopy to measure the binding of doxorubicin to chromatin. As the authors point out, the spectral overlap between GFP and doxorubicin can complicate this measurement. For the measurements described Fig 2c, the authors have chosen detection wavelengths with minimal spectral overlap. Importantly, in Fig 2C the authors present experimental evidence that potential spectral overlap did not affect the measurement by showing that doxorubicin did not change the fluorescence lifetime of free GFP. For endomicroscopy the detection wavelength had to be changed due to reflection signal. However, this may also lead to a potential mixture of the signals from H2B-GFP and doxorubicin lifetime. The authors acknowledge this problem and developed multiple modelling approaches to obtain clean GFP fluorescence lifetimes. Nevertheless, real experimental evidence is lacking. Can the authors add an experimental control in Figure 2F and Supplementary Figure 2 in which they measure the fluorescence lifetime of free GFP (not tagged on histones) at various doxorubicin (similar to the experiments performed in Figure 2c)?

- The authors used their developed FLIM endomicroscopy to investigate the intra- and inter-tumor heterogeneity in chromatin binding by doxorubicin. Since the authors only present static images of single time points per mouse, all presented measurements could also have been performed by ex vivo analysis of isolated tumors using conventional FLIM systems (e.g. using the scope used to obtain data in Fig 1 and Fig2a-c). To make this manuscript more compelling, it will be important to show the advances of this technique over the currently available FLIM techniques. For example, the authors can perform repeated intravital microscopy experiments of the same tumors at various time points after doxorubicin treatment. It would be interesting to test whether variations of doxorubicin engagement lead to different tumor regression outcomes which could only be tested by imaging the same nodules at various time points.
- Figure 4 illustrates a striking inefficiency of doxorubicin delivery by the intravenous route 3 hours after IV administration. However, did the authors ever check later time points? Would it be possible that the arrival of doxorubicin after IV injection is delayed? Can the authors include measurements at later time points (e.g. 24, 48 hours for both IP and IV administration)?
- In Figure 4, can the authors include the images of the nodules that were analyzed? Moreover, can the authors describe the number of nodules that were measured per mouse?
- The authors observed large variations in doxorubicin engagement between IV administration and IP administration. Does the variation in doxorubicin engagement also correlate with the response? Does IP administration of doxorubicin but not IV administration lead to tumor regression?

Minor comments:

- A large part of the manuscript describes the development of their technique. The title of the manuscript does not represent this part of the manuscript.
- Figure 2b – The color legend is not visible.
- Figure 2d and e – The images are clear, but not very informative. A quantification of the data next to the images would improve the figure.
- Line 220 – 249 are lengthy and could be shortened by moving the details to the methods sections. Moreover the corresponding Figure 3 does not contain any data, therefore I would recommend to move it to the Supplementary Figures.
- Figure 6a – In the text it is stated that 0.9 μM doxorubicin reduces the cell viability by almost 90% (line 318). This does not seem to correspond with the figure, where a reduction of about 65% is shown.

Reviewer #1 (Remarks to the Author):

This is a very comprehensive and nicely written manuscript addressing an important problem in cancer therapy, i.e. the efficacy of drug delivery and drug-target engagement. The authors present a systematic approach to imaging (both in vitro and in vivo) of doxorubicin binding to chromatin using fluorescence lifetime imaging confocal endomicroscopy. The imaging results are significant while intriguing.

We are delighted that the reviewer found our study ‘very comprehensive’, ‘nicely written’, and that it addressed ‘an important problem in cancer therapy’. We also thank him/her for their careful reading of the work and thoughtful questions.

Response to points of clarification

1. A 80 MHz rep rate laser was used in the study. At such high rep rate the sample/fluorophores can be photobleached. This aspect needs to be addressed as it could affect the validity of the results.

Response The reviewer raises an important point. We did not observe photobleaching during the acquisition period. To address this quantitatively, we have now added analysis in Supplementary Figure 3b showing that the mean pixel count does not diminish during the FLIM acquisition.

2. A method that allows for tracking the motion of the tumor in vivo was used to account for the small number of photons captured per frame. The shortcomings of this method need to be discussed as they might affect the results from the in vivo measurements. This could contribute to the perceived/reported heterogeneity in doxorubicin chromatin binding.

Response The reviewer is correct that we did not discuss in detail the motion correction part of processing of the CM FLIM data or its shortcomings. This is a rigid body registration and hence cannot deal well with shear or other complex deformations of tissue. We should therefore consider the possibility that there will be some incorrect assignment of FLIM data pixels to the wrong cells. While this would be problematic if we were attempting to discern sub-cellular spatial variations in fluorescence lifetime, here we are only calculating the mean fluorescence lifetime per nucleus and so we only need to be confident that the extent of any misalignment is less than the distance between nuclei and therefore the risk of the motion tracking algorithm mis-assigning a pixel and its associated lifetime information to the wrong nucleus is very low. To address this issue, we have added the following text to the Materials and methods section to provide a brief an explanation of the motion correction data processing:

For live FLIM updates, FIFO acquisition mode was used and in-house written software, including a graphical user interface (GUI), was used to compile photon events into an image at an update rate of 8.5 Hz. For each measurement, at least 30 seconds of FLIM data was recorded and a subset of

85 frames (corresponding to a 10 second FIFO TCSPC acquisition) were aligned and used for subsequent FLIM analysis, as depicted in figure 3. To align these frames, a sub-region of the first frame was selected to serve as a template for alignment of the following frames relative to this first “reference frame”. The lateral shift between frames was determined from the location of the peak of the normalized image cross-correlation (calculated using the OpenCV C++ template-matching function ‘matchTemplateOCV.cpp’ available in MATLAB). Frames having a low normalized cross-correlation, e.g. during breathing were rejected. The sequences of 85 image frames were selected on the basis of their average normalized cross-correlation value. Typically, frames with a value above 0.9 were retained for subsequent FLIM analysis, but in some cases (usually those with lower signal to noise ratio) a cut-off of down to 0.8 was used. Frames having a low normalized cross-correlation, e.g. those significantly distorted due to breathing, did not contribute to the FLIM analysis and the image distortion over each sequence of the 85 selected frames was observed to be much less than the inter-nuclear distance. Thus, the risk of erroneously mixing photon arrival times between nuclei in the calculation of the mean lifetimes of each cell nucleus was low.

Thus, any frames for which the image deformations are severe would be discarded and would not contribute to the calculation of fluorescence lifetimes in the cell nuclei. For the reviewer’s benefit, we present below images of the first reference frame and the superposition of all the sequences in the frame for two exemplar sequences with threshold normalized cross-correlation values of 0.9 and 0.82. Both data sets illustrate that the frame alignment has worked well and even for frame sequences with a normalized cross-correlation threshold of 0.82, any relative displacement of different cell nuclei is small compared to their separation. Thus, we can be confident that our motion correction has been sufficiently successful to avoid cross-talk artefacts in the calculation of the mean lifetimes per cell nucleus.

Panel shows frames from two exemplar FLIM data sets obtained using threshold normalized cross-correlation values of (a) 0.9 and (b) 0.8: For each data set, the top left image is the first “reference” frame, the top right is the sum of all the aligned frames, the middle row shows expanded images of the sub regions used for the template matching (as indicated in green frames on top row). The bottom left image is the two template regions superimposed in green and magenta and the bottom right image shows the line profiles for the reference image (blue) and summed image (green) sequence and which corresponding to the dashed red lines indicated in the middle right images.

3. The cellular metabolism in vivo (animals) differs from the in vitro (cell culture). Does that affect the doxorubicin-chromatin engagement and potentially contribute to the discrepancies observed between doxorubicin effects in vitro vs in vivo?

Response The reviewer is quite right that there are significant differences in metabolic state between in vitro experiments and the in vivo situation. We now include a mention of this on lines 351 and 352 of the Discussion:

“Inter-tumor and inter-nodule variation may reflect localized differences in the metabolic state and rate of drug uptake, although differences in doxorubicin binding observed in adjacent cells would be difficult to explain in terms of localised differences in metabolism alone. Particularly dense extracellular matrix may also act as a local barrier to drug access.”

We have also run a simple experiment in vitro to compare DOX binding to chromatin in varying glucose conditions (either 25mM or 5mM), see figure below. This indicates that 5-fold variations in

extracellular glucose do not affect chromatin binding by doxorubicin. An exhaustive study of this issue would be an entire study in its own right and, as we comment in the Discussion, we think it unlikely that there are sufficiently dramatic local differences in nutrient availability for metabolism alone to account for the highly localized variation in doxorubicin binding, such as between adjacent cells.

Reviewer #2 (Remarks to the Author):

In their manuscript, Sparks and colleagues describe the development of a novel quantitative microscopy approach based on FLIM using a confocal endomicroscope to visualize and quantify drug-target engagement in vivo. In the first part of the manuscript the technique is tested and validated. In the second part the authors show that the confocal endomicroscopy technique can be used to visualize the binding of doxorubicin to chromatin in a model of metastatic ovarian cancer. In this model, they reveal a large intra- and inter-tumor heterogeneity in chromatin binding by doxorubicin. Moreover, the data shows that the mode of drug administration, i.e. intra-peritoneal (IP) or intra-venous (IV), leads to a striking variation in drug delivery in this tumor model.

The development of the quantitative confocal endomicroscopy is an impressive effort and the presented biological findings are very interesting. Nevertheless, the authors need to address my major points before I can recommend to publish this manuscript at Nat Commun.

Response We are pleased that the reviewer found our study ‘an impressive effort’ and that the ‘presented biological findings are very interesting’. We also note several important points that he/she raised, and we address these in turn below.

Response to major comments

- 1 In the first part of the manuscript the authors describe how they have used FLIM endomicroscopy to measure the binding of doxorubicin to chromatin. As the authors point out, the spectral overlap between GFP and doxorubicin can complicate this measurement. For the measurements described Fig 2c, the authors have chosen detection wavelengths with minimal spectral overlap. Importantly, in Fig 2C the authors present experimental evidence that potential spectral overlap did not affect the measurement by showing that doxorubicin did not change the fluorescence lifetime of free GFP. For endomicroscopy the detection wavelength had to be changed due to reflection signal. However, this may also lead to a potential mixture of the signals from H2B-GFP and doxorubicin lifetime. The authors acknowledge this problem and developed multiple modelling approaches to obtain clean GFP fluorescence lifetimes. Nevertheless, real experimental evidence is lacking. Can the authors add an experimental control in Figure 2F and Supplementary Figure 2 in which they measure the fluorescence lifetime of free GFP (not tagged on histones) at various doxorubicin (similar to the experiments performed in Figure 2c)?

Response The reviewer raises an important point about validating our model linking fluorescence lifetime to doxorubicin concentration. We have now addressed this experimentally by performing imaging of free GFP using the endomicroscope. The revised panels in Figure 2f and Supplementary Figure 2f now show that the observed drop in the fluorescence lifetime of free GFP for increasing concentration of doxorubicin closely matches the model – note the green line that shows the observed drop in the measured lifetime of free GFP and how closely this matches the red line showing the expectation based on our model. This gives us further confidence in our ability to relate changes in the fluorescence of lifetime of H1-GFP to doxorubicin concentration without erroneous signal from direct doxorubicin emission.

- 2 The authors used their developed FLIM endomicroscopy to investigate the intra- and inter-tumor heterogeneity in chromatin binding by doxorubicin. Since the authors only present static images of single time points per mouse, all presented measurements could also have been performed by ex vivo analysis of isolated tumors using conventional FLIM systems (e.g. using the scope used to obtain data in Fig 1 and Fig2a-c). To make this manuscript more compelling, it will be important to show the advances of this technique over the currently available FLIM techniques. For example, the authors can perform repeated intravital microscopy experiments of the same tumors at various time points after doxorubicin treatment. It would be interesting to test whether variations of doxorubicin engagement lead to different tumor regression outcomes which could only be tested by imaging the same nodules at various time points.

Response The reviewer argues that longitudinal imaging of the same tumor nodule will provide more compelling data than imaging of different nodules at different times. We have now carried out this analysis and added the new data to Supplementary Figures 4 & 5. Briefly, we imaged the same nodules at 45, 90, and 180 minutes after intraperitoneal delivery of doxorubicin. In accordance with our procedural license, the mouse was maintained under anesthesia for the duration of the imaging session. The small nature of the incision required for the use of device meant that the physiological state of the mouse was easily maintained with the incision closed in the period between image acquisition to prevent dehydration. As is now shown in Supplementary Figures 4 & 5, the highest drug target

engagement was at 45 and 90 minutes post intraperitoneal injection and, in all cases, was lower at 180 minutes. These data are consistent with the pattern observed across different nodules in different mice at 90 and 180 minutes. We believe that these new data make a strong addition to the study and thank the reviewer for suggesting them.

In the longer term, we would like to perform these type of analyses over many days in the same animal and try to correlate drug target engagement with the ultimate cell fate. However, this was not possible in this context as doxorubicin has little effect on cell viability over 48 hours (Figure 6) and without at least a partial response (i.e. a significant fraction of the cells dying) there is nothing to which variable drug binding can be correlated. We hope to perform this type of experiment in the future in syngeneic ovarian cancer models in which we can also visualize drug binding to stromal cells. This would be a new study in its own right and also require revisions to our procedural license. Nonetheless, we completely agree with the reviewer regarding the necessary 'direction of travel'.

To address the Reviewer's point in the manuscript, we have added the following new text to the Results section:

"A key motivation for developing a confocal endomicroscope capable of fluorescence lifetime imaging was to enable repeated imaging of the same tumor nodules with minimal tissue disruption. We therefore sought to perform repeat imaging on the same nodule at different times following delivery of doxorubicin. Supplementary Figure 4b shows fluorescence lifetime measurements of three different nodules at 45, 90, and 180 minutes after intra-peritoneal injection of 5mg/kg doxorubicin. Between image acquisition, the peritoneal cavity was kept closed to avoid dehydration. Consistent with the observations across different mice (Figure 4), doxorubicin – chromatin engagement declined between 90 and 180 minutes. These data demonstrate the capability for longitudinal imaging of tumors."

- 3 Figure 4 illustrates a striking inefficiency of doxorubicin delivery by the intravenous route 3 hours after IV administration. However, did the authors ever check later time points? Would it be possible that the arrival of doxorubicin after IV injection is delayed? Can the authors include measurements at later time points (e.g. 24, 48 hours for both IP and IV administration)?

Response We have now tested this by performing imaging after 24 hours of mice treated with intravenous doxorubicin. As shown in the revised version of Figure 4 and Supplementary Figure 4 & 5, we were unable to see any significant engagement of doxorubicin with chromatin after 24 hours. This argues against the slow delivery of doxorubicin to the tumor via the intravenous route. Given the lack of signal following drug administration by both routes at 24 hours, we did not pursue analysis at 48 hours. Further, conventional pharmacokinetic analysis indicates that it is very unlikely to have a gain in drug access after 24 hours (Johansen - Cancer Chemother Pharmacol. 1981;5(4):267-70).

- 4 In Figure 4, can the authors include the images of the nodules that were analyzed? Moreover, can the authors describe the number of nodules that were measured per mouse?

Response The reviewer makes two good suggestions. We have now added to Figure 3 to make it visually clearer how the device is used and what a nodule growing on the intestine and into the peritoneal wall looks like. Further, we have added Supplementary Table 2 making it clear how many mice and nodules were imaged throughout the study.

- 5 The authors observed large variations in doxorubicin engagement between IV administration and IP administration. Does the variation in doxorubicin engagement also correlate with the response? Does IP administration of doxorubicin but not IV administration lead to tumor regression?

Response We agree with the logic of the reviewer's comment, however the lack of response to even intraperitoneal delivery of doxorubicin that is shown in Figure 6b make it unlikely that intravenous delivery will yield any response. Nonetheless, we performed the intravenous delivery experiment that was requested and have added the data to Figure 6b. In line with our expectation, there was little effect of doxorubicin 48 hours after intravenous delivery, if anything the tumors were larger.

New figure 6b showing bioluminescence images of nude mice bearing IGROV1-luciferase tumors (upper panels) immediately prior to IP or IV delivery of doxorubicin (5mg/kg) and (lower panels) show the same mice imaged 48 hours later.

Minor comments:

- A large part of the manuscript describes the development of their technique. The title of the manuscript does not represent this part of the manuscript.

Response We agree and have changed the title to ‘Development of in vivo fluorescence lifetime imaging confocal endomicroscopy reveals intra- and inter-tumor heterogeneity in chromatin binding by doxorubicin’.

- Figure 2b – The color legend is not visible.

Response We have now made the legend wider so that it can easily be seen.

- Figure 2d and e – The images are clear, but not very informative. A quantification of the data next to the images would improve the figure.

Response Figure 2f provides the quantification of Figures 2d, 2e and related FLIM data. We have modified the text to make this clearer.

- Line 220 – 249 are lengthy and could be shortened by moving the details to the methods sections. Moreover the corresponding Figure 3 does not contain any data, therefore I would recommend to move it to the Supplementary Figures.

Response Even though the Figure does not contain key experimental data, it is central to the explaining the procedure used throughout the study and therefore we would like to keep it as a main figure. We have also added images of tumor nodules as the reviewer requested. If the reviewer and editor feel strongly about moving this figure to the Supplementary Figures, then we would oblige. We have removed two sentences that were somewhat repetitive with the Introduction to shorten the section.

- Figure 6a – In the text it is stated that 0.9 μM doxorubicin reduces the cell viability by almost 90% (line 318). This does not seem to correspond with the figure, where a reduction of about 65% is shown.

Response The reviewer is quite right, we apologize for this mistake and we have now corrected it. Our overall point about the contrasting efficacy of doxorubicin *in vitro* vs *in vivo* remains strong.

We also changed reference 25 to a more authoritative paper from which the replaced paper was derived.

REVIEWERS' COMMENTS:

Reviewer #1 (Remarks to the Author):

In the revised manuscript, the authors have addressed all major concerns. I have no additional comments.

Reviewer #2 (Remarks to the Author):

The authors have done an excellent job and have addressed my major concerns. It is unfortunate that doxorubicin has little effect on cell viability over 48 hours, and that it did not make sense to perform multi-day imaging. However, the new data of multiple imaging sessions at 45, 90, and 180 minutes also illustrates the advances of this technique over the currently available static FLIM techniques, and is therefore a good alternative to address my concern. Moreover, it is good to hear that the authors will perform multi-day imaging experiment in syngeneic ovarian cancer models in the future, and I agree that this will be a new study in its own right and that the development and imaging of a new cancer model goes beyond the scope of this manuscript. All in all, the authors have presented new data that have addressed all my major concerns and now I support publishing this important work in Nat Comm.